# Placement Method of Multiple Lidars for Roadside Infrastructure in Urban Environments

**DOI:** 10.3390/s23218808

**Published:** 2023-10-29

**Authors:** Tae-Hyeong Kim, Gi-Hwan Jo, Hyeong-Seok Yun, Kyung-Su Yun, Tae-Hyoung Park

**Affiliations:** 1Research and Development Department, Korea Intelligent Automotive Parts Promotion Institute (KIAPI), Daegu 43011, Republic of Korea; thkim@kiapi.or.kr (T.-H.K.); gudtjr0124@kiapi.or.kr (H.-S.Y.); kadbonow@kiapi.or.kr (K.-S.Y.); 2Department of Control and Robot Engineering, Chungbuk National University, Cheongju 28644, Republic of Korea; ghjo@chungbuk.ac.kr; 3Department of Intelligent Systems and Robotics, Chungbuk National University, Cheongju 28644, Republic of Korea

**Keywords:** placement, optimization, genetic algorithm, sensors, infrastructure, urban intersection

## Abstract

Sensors on autonomous vehicles have inherent physical constraints. To address these limitations, several studies have been conducted to enhance sensing capabilities by establishing wireless communication between infrastructure and autonomous vehicles. Various sensors are strategically positioned within the road infrastructure, providing essential sensory data to these vehicles. The primary challenge lies in sensor placement, as it necessitates identifying optimal locations that minimize blind spots while maximizing the sensor’s coverage area. Therefore, to solve this problem, a method for positioning multiple sensor systems in road infrastructure is proposed. By introducing a voxel grid, the problem is formulated as an optimization challenge, and a genetic algorithm is employed to find a solution. Experimental findings using lidar sensors are presented to demonstrate the efficacy of this proposed approach.

## 1. Introduction

In the realm of autonomous driving research, sensors play a pivotal role. Cameras, for instance, perceive incoming light through their lenses and image sensors. Radar, short for “radio detection and ranging”, employs electromagnetic waves to estimate position and relative velocity.

Lidar, or “light detection and ranging”, stands out as a key sensor in autonomous driving research. It gauges position by measuring the time it takes for a laser beam to be emitted, reflect off an object, and return. Lidar, especially the time-of-flight (TOF) variant, is widely used in autonomous vehicles to acquire point cloud data, aiding in object detection, localization, and map construction. Integrating lidar into roadside infrastructure is emerging, and some studies have been conducted using roadside lidar [1,2,3,4,5,6,7,8,9,10].

A high-channel lidar, typically having 64 channels or more, offers advantages such as narrower beam spacing, facilitating feature extraction and the detection of distant objects. However, these sensors are expensive, making it difficult to install multiple units in one space. Moreover, they are constrained by their limited measurement range and susceptibility to occlusion.

Research on the placement of multiple lidars has primarily focused on vehicles. S. Roos et al. [11] placed multiple lidars and assessed their performance using the CARLA simulator. T. Kim et al. [12] introduced a multi-lidar placement approach that takes into account blind spots created by other vehicles. They placed a 2D occupancy grid board at a specified distance and calculated occupancy using the point cloud reflected on the board. Furthermore, various studies have proposed techniques for placing multiple sensors within a single vehicle [13,14,15,16,17].

Existing research on lidar placement primarily focuses on vehicles and does not consider the placement problem for infrastructure sensors. Urban environments often feature numerous obstructions, such as streetlights and traffic signals, at intersections. These obstructions can lead to a decline in the quality of raw data due to occlusion. Figure 1 shows a point cloud acquired from a typical intersection. The quality of these raw data varies depending on lidar placement, leading to a drop in object detection performance. Therefore, the optimization of sensor placement in urban settings emerges as a crucial challenge. Infrastructure lidar placement offers a higher degree of freedom, requiring consideration of both the xyz positions and roll, pitch, and yaw angles [18]. X. Cai et al. [18] conducted a study where they placed multiple lidars at an intersection, analyzing their impact on recognition performance based on their placements. S. Jin et al. [19] put forth an evaluation method for lidar placement, but it lacks a systematic approach, including accounting for blind spots in modeling the real environment.

A. Qu et al. [20] proposed implementing the environment in a simulation and placing sensors, designating candidate locations, and optimizing them using the point cloud projected on the road surface while excluding buildings, sidewalks, etc. L. Kloeker et al. [21] proposed optimizing lidar placement in infrastructure by utilizing a 3D digital map. They divided the road into triangles based on the OpenDRIVE [22] format map and optimized lidar placement by considering the number of points projected onto these triangles.

Several studies have used multiple lidars. However, they often fail to reflect the unique features of urban roads, characterized by numerous buildings, poles, and more, or have primarily relied on simulations. In this study, we propose a method for multiple sensors placement, aiming to identify the optimal position and orientation that maximizes data detection range. Our proposed method takes into account blind spots arising from the road environment. To quantify parameters for sensor placement, we introduce a novel approach for evaluating occupancy through voxelization of a map. Additionally, we employ a genetic algorithm to address the sensor placement problem. The sensor array is represented by chromosomes, and new chromosomes are generated through crossover and mutation operators. The experiments of this study were conducted in a simulation replicating the real environment, and the results affirm the effectiveness of our proposed method. The key contributions of this paper are as follows:
Modeling the lidar placement environment based on real-world data rather than simulation.Converting the point cloud map and lidar beams into a computable discrete signal format (voxel) for quantitative evaluation.Optimizing the positions and directions of the lidars using genetic algorithm chromosomes and introducing a 2-opt local optimization method.Proposing a placement method for multiple lidars (two or more) on infrastructure, replicating and validating the simulation placement results in a real environment.

This paper is structured as follows. Section 2 introduces a multiple lidar system in an urban environment, and Section 3 provides a mathematical description of the problem. Section 4 describes the optimization algorithm, while Section 5 presents the experimental results.

## 2. Multiple Lidar System in an Urban Environment

The mechanically rotating 3D lidar radiates $N_{bm,\;\theta }$ channel laser beams in the vertical direction, forming a rotating laser beam array in the horizontal direction. The angle between these beams in the vertical or horizontal direction is referred to as the resolution, denoted by $∆{ang}_{bm,\;\theta }^{\alpha }\;(\alpha =1,\;\ldots ,\;N_{bm,\;\theta })$ and $∆{ang}_{bm,\;\phi }$, respectively. The lowest and highest angles in the vertical direction are represented as $ang_{bm,\;\theta }^{-}$, and $ang_{bm,\;\theta }^{+}$, respectively.

The lidar sensor calculates distances by measuring the time taken for emitted laser beams to reflect off objects and return. This information is represented as points. Figure 2 shows the resolution and reflection points of the lidar.

When a lidar is placed along an urban road, it may encounter obstacles like buildings that can block the laser beams, causing what is commonly known as a “dead zone”. To minimize dead zones, multiple lidars are strategically positioned at intersections. Each lidar covers a specific area, resulting in overlapping coverage regions. Figure 3 provides an example of multiple lidar systems, their coverage areas, and the overlapping coverage regions. Figure 3a displays the positions of three lidars placed at an intersection, with the cyan polygons representing buildings that replicate the urban environment and create blind spots in sensing. Figure 3b shows the coverage of each lidar in relation to the blind spots caused by the buildings. Figure 3c demonstrates the overlapping coverage provided by the three lidars, effectively reducing blind spots at intersections through their strategic deployment. Almost all areas around intersections are covered by lidar.

In a multiple lidar system containing $N_{ld}$ lidars, the position of the l-th lidar is as follows.
(1)$$p_{ld}^{l}=\left(x_{ld}^{l},y_{ld}^{l},z_{ld}^{l},\;r_{ld}^{l},\;p_{ld}^{l}\right)\in S(l=1,\cdots ,N_{ld})$$
where S is the set of potential positions for the placement of lidars. The position ($p_{ld,\;1}$) of the first (reference) lidar is fixed by the user, while the positions of the remaining lidars are determined using the proposed method.

The number and density of lidar points increase in areas where the coverage overlaps. Efficient placement of multiple lidars reduces dead zones at urban intersections, as shown in Figure 4, and enables the acquisition of high-resolution data, similar to that achievable with high-channel lidars.

## 3. Problem Formulation

The main problem of multiple lidar placement in urban environments is minimizing blind spots while maximizing point density. One of the main problems is that the locations for lidar placement are restricted to specific areas, such as the roadside, to avoid interfering with the road where vehicles are in operation. Therefore, an optimization method is required to identify the optimal placement while modeling the real environment to reduce blind spots.

In this study, we introduce the concept of Lidar Occupancy Space (LOS) for optimization. LOS comprises voxels, which are a down-sampling method that reduce the number of point clouds and converts them into a normalized discrete signal format. This enables the modeling of the real environment for computational purposes and reduces the computational load for layout optimization. An LOS is generated to match the size of the user’s region of interest $\left({LOS}_{x},\;{LOS}_{y},\;{LOS}_{z}\right)$. Since the point cloud map, derived from real lidar data, accurately reflects the actual environment, the LOS is weighted using this map, and the occupancy rate is determined as the lidar beams traverse the LOS.

Additionally, we propose a Lidar Occupancy Voxel (LOV) grid to assess the distribution of lidar beams. LOV consists of unit-length cubes ($U_{LOV}$). The LOS is divided into $W_{LOS}\times D_{LOS}\times H_{LOS}$ voxel grids, which are determined according to the size of the LOS and the unit length ($L_{LOV}$) of the LOV. Figure 5 shows LOV and LOS, where LOS is an aggregation of LOVs.

To ensure the accurate weighting of LOS using the point cloud map, the point cloud map must undergo preprocessing (filtering) to match the size of (${LOS}_{x},\;{LOS}_{y},\;{LOS}_{z}$). When the LOS and point cloud map overlap, the distribution of points within each LOV will vary, as shown in Figure 6. Figure 6a presents the point cloud map, while Figure 6b shows the LOS generated from this map. The weight of the LOV comprising the LOS is determined by the number of points contained within the corresponding voxel, with yellow indicating the presence of weight. For the voxel grid LOV(i, j, k), the weight ($LOV_{wg}(i,\;j,\;k))$ is defined as follows.
(2)$${LOV}_{wg}\left(i,\;j,\;k\right)=\left\{\begin{array}{cc} 1 & ,\;N_{wg}(i,j,\;k)\geq 1\\ 0 & ,\;N_{wg}\left(i,j,\;k\right)=0\; \end{array}\right.$$
where $N_{wg}(i,j,\;k)$ is the number of points of the point cloud map included in $LOV\left(i,\;j,\;k\right)$. ${LOV}_{wg}\left(i,\;j,\;k\right)$ denotes whether the grid is occupied by the point cloud map, and has a weight if it is occupied.

In the proposed method, occupancy is evaluated based on whether the lidar beam passes through the voxel grid $LOV\left(i,\;j,\;k\right)$ that constitutes the LOS, and is determined as follows:
**Step** **1.** Set lidar counter l to 1.**Step** **2.** Set the horizontal angle counter β to 0.**Step** **3.** Set the vertical angle counter α to 1, and set the vertical angle variable angbm, θ^ to $ang_{bm,\;\theta }^{-}$.**Step** **4.** Calculate the intersection $p_{bm}(\alpha ,\;\beta )$ of the six outermost surfaces of the LOS and the lidar beam bm(α, β) as follows.
(3)xbmα, β=rcos⁡angbm, θ^cos⁡(β∆angbm, ϕ)
(4)ybmα, β=rcos⁡angbm, θ^sin⁡(β∆angbm, ϕ)
(5)zbmα, β=rsin⁡angbm, θ^
where $r=∥p_{bm}\left(\alpha ,\;\beta \right)-p_{ld}^{l}∥$ is the distance between the lidar and the intersection.**Step** **5.** Using the Bresenham algorithm [23], store index $idx^{\gamma }=(i,\;j,\;k)$ of the voxel grid $LOV\left(i,\;j,\;k\right)$ included in the line segment between $p_{ld}^{l}$ and $p_{bm}(\alpha ,\;\beta )$ in the array IDX.
(6)$$IDX=\{idx^{0},\;\ldots ,\;idx^{N_{IDX}}\}$$
where $N_{IDX}$ is the number of voxel grids included in the line segment. The Bresenham algorithm is a computer graphics algorithm designed for drawing straight lines using integer calculations exclusively, avoiding the complexity and slowness associated with real number calculations. Since actual computer screens consist of pixels, and pixels are inherently integers, a straight line drawn through a straight-line equation may span multiple pixels. In this study, we generated LOS by introducing voxels and efficiently approximated straight lines within the LOV that constitutes the LOS.**Step** **6.** Set the index search counter γ to 0.**Step** **7.** If ${LOV}_{wg}\left(idx^{\gamma }\right)=1$, change $LOV\left(idx^{0}\right)$ to ${LOV}_{wg}\left(idx^{\gamma }\right)$ to 1 and move to Step 9.**Step** **8.** Increment the index search counter γ and move to Step 7.**Step** **9.** Increase the vertical angle counter α and calculate angbm, θ^ as in the following equation. If angbm, θ^ ≤ angbm, θ+, move to Step 4.
(7)angbm, θ^=angbm, θ^+angbm, θα**Step** **10.** Increase the horizontal angle counter β. If $\beta \;\leq \;\frac{2\pi }{∆ang_{bm,\;\phi }}$, move to step 3.**Step** **11.** Increase lidar counter l, If $l\;\leq N_{ld}$, move to Step 2.

The total lidar occupancy LO of lidar beams relative to the LOV, as shown in Figure 7, is the sum of all voxel grids, and is calculated as follows.
(8)$$LO=\sum_{k=1}^{H_{LOS}}\sum_{j=1}^{D_{LOS}}\sum_{i=1}^{W_{LOS}}LOV(i,\;j,\;k)$$

On the other hand, lidar occupancy %LO is the ratio of LO to the total number of voxel grids.
(9)$$\%LO=\frac{LO}{W_{LOS}\times D_{LOS}\times H_{LOS}}$$

The lidar placement problem aims to determine the lidar locations $p_{ld}^{l}\;\in S(l=1,\cdots ,N_{ld})$ to maximize the lidar occupancy LO. The voxel grid value depends on the lidar’s placement, and the mathematical formula is defined as follows.
(10)$$p_{ld}^{l,\;*}=\underset{p_{\mathrm{ld}}^{l}\;\in S}{\mathrm{argmax}}\sum_{k=1}^{H_{LOS}}\sum_{j=1}^{D_{LOS}}\sum_{i=1}^{W_{LOS}}LOV(i,\;j,\;k)$$

## 4. Optimization

This section describes the placement optimization algorithm. The placement algorithm is designed to find a near-optimal solution. The genetic algorithm is a search method that identifies the optimal solution by simulating the way organisms evolve and adapt to their environment. This algorithm operates by selecting the chromosome with the best fitness from a set of chromosomes and iteratively refining the search in the direction of the optimal solution. The sensor placement algorithm is as follows:
**Step** **1.** Place the first (reference) lidar.**Step** **2.** Create lidar occupancy space (LOS) using the point cloud map.**Step** **3.** Assign weights $LOV_{ws}(i,\;j,\;k)$ to the voxel grid.**Step** **4.** Set the lidar count l to 2.**Step** **5.** Find the $p_{ld}^{l,\;*}$ that maximizes the lidar occupancy (LO) using a genetic algorithm.**Step** **6.** Increment the lidar counter. If $l\;\leq N_{ld}$ go to Step 5.

In Step 5, a genetic algorithm (GA) is applied to determine the placement of the sensors. The chromosome pop is a binary string divided by five sections as:(11)$$pop=<X_{pop},\;Y_{pop},\;Z_{pop},\;R_{pop},\;P_{pop}>$$
(12)$$X_{pop}=<s_{x}^{1},\;s_{x}^{2},\;\ldots ,\;s_{x}^{N_{x}}>,\;s_{x}\in \;\{0,1\}$$
(13)$$Y_{pop}=<s_{y}^{1},\;s_{y}^{2},\;\ldots ,\;s_{y}^{N_{y}}>,\;s_{y}\in \;\{0,1\}$$
(14)$$Z_{pop}=<s_{z}^{1},\;s_{z}^{2},\;\ldots ,\;s_{z}^{N_{z}}>,\;s_{z}\in \;\{0,1\}$$
(15)$$R_{pop}=<s_{r}^{1},\;s_{r}^{2},\;\ldots ,\;s_{r}^{N_{r}}>,\;s_{r}\in \;\{0,1\}$$
(16)$$P_{pop}=<s_{p}^{1},\;s_{p}^{2},\;\ldots ,\;s_{p}^{N_{p}}>,\;s_{p}\in \;\{0,1\}$$
where $X_{pop},\;Y_{pop},\;Z_{pop},\;R_{pop}$, and $P_{pop}$ are matched with the lidar placement $p_{ld}^{l}=\left(x_{ld}^{l},y_{ld}^{l},z_{ld}^{l},r_{ld}^{l},p_{ld}^{l}\right)$, respectively.

The initial population, denoted as $POP^{g}=<pop^{g,\;1},\;pop^{g,\;2},\;\ldots ,\;pop^{g,\;N_{POP}}>$, is generated through random number generation. The next population is formed by the selection operator, with the remaining stochastic sampling [24] reproduces to chromosomes with higher fitness, where fitness is defined as the total lidar occupancy LO.

The crossover and mutation operators create a range of new chromosomes, enhancing the optimization of lidar placements. Chromosomes are randomly selected based on the crossover probability (${PB}_{cs}$), and the crossover point is randomly determined at the boundary of sections, as showed in Figure 8. New chromosomes $\bar{pop^{g,\;1}}$ and $\bar{pop^{g,\;2}}$ are generated through the crossover operation of $pop^{g,\;1}$ and $pop^{g,\;2}$. Similarly, chromosomes are randomly selected according to the mutation probability (${PB}_{mt}$). For chromosome $\bar{pop^{g,\;1}}$, one bit from each section ($X_{pop},\;Y_{pop},\;Z_{pop},\;R_{pop},\;P_{pop})$ is selected. A new chromosome $\overset{̿}{pop^{g,\;1}}$ is generated by inverting the selected bits, as depicted in Figure 9.

Our genetic algorithm is summarized as follows:
**Step** **5-1.** Generate initial population $POP^{0}=<pop^{0,\;1},\;pop^{0,\;2},\;\ldots ,\;pop^{0,\;N_{POP}}>$ randomly.**Step** **5-2.** Set the generation counter p to 1.**Step** **5-3.** Calculate the fitness, and reproduce chromosomes by the remainder stochastic sampling.**Step** **5-4.** Pairs of chromosomes are randomly selected, and crossover operation is performed between chromosomes.**Step** **5-5.** Chromosomes are randomly selected, and mutation operation is performed.**Step** **5-6.** Find the best chromosome and vary it by 2-opt improvement [25]. The 2-opt method is a local search algorithm that examines all feasible combinations and swaps them to find a solution.**Step** **5-7.** Make increments to the generation counter p. If the exit condition is not satisfied, go to Step 5-3.**Step** **5-8.** The final best chromosome is decoded into placement of lidar position $p_{ld}^{l,\;*}$.

## 5. Experiments

### 5.1. Experimental Setup

The proposed sensor placement algorithm was validated using 3D lidars. Two or three forty-channel lidars (HESAI Pandar40P, Shanghai, China) were simulated and used for placement optimization. For performance comparison, a single 128-channel lidar (Velodyne VLS-128, San Jose, CA, USA) was utilized. The algorithm was executed on a workstation running Linux Ubuntu 20.04 and ROS Noetic, with the program developed in the C++ language. The specifications of the lidars used in the experiment are detailed in Table 1.

The test area selected for the experiment was the proving ground of the Korea Automotive Parts Promotion Institute (KIAPI). KIAPI’s proving ground encompasses various test facilities, including an autonomous vehicle test road, a multipurpose test track, and a high-speed circuit. This experiment specifically focused on the autonomous vehicle test road, which features two four-way intersections. These intersections replicate an urban environment and exhibit blind spots created by buildings, as shown in Figure 10.

Table 2 provides a comprehensive overview of the experiment. As part of the simulation, the placement optimization results of two or three forty-channel lidars and one one-hundred-and-twenty-eight-channel lidar were compared at Intersection #1 (S1~S3) and Intersection #2 (S4~S6). The optimization of lidar placement involved adjusting the crossover and mutation probabilities.

The results of the optimal placement in the simulator at Intersection #1 were subsequently replicated in a real environment (R1~R6). In the real environment, the vehicle was driven, and its detection was based on data acquired from the placed lidars, with a comparison of detection ranges (R1~R2). Additionally, pedestrians walking in the real environment were detected using data from the lidars, and their detection ranges were compared (R3~R4). Finally, when two vehicles were driven, one vehicle was partially obscured, and the detection range was compared by detecting the vehicles from the data acquired by the placed lidars. To place lidars on a simulator using the proposed method and verify their performance, an experiment was conducted at two intersections with different environments. Furthermore, by optimally placing multiple 40-channel lidars, which is relatively cost effective, it was intended to reduce the blind spot of the intersection and maintain the detection performance. We sought to validate the effectiveness of the proposed method by applying the simulation results to a real environment. We aimed to verify the utility of the proposed placement method by evaluating the detection performance of objects primarily found in infrastructure, such as vehicles and pedestrians, and demonstrating its reproducibility.

### 5.2. Experimental Results

#### 5.2.1. Placement Optimization Simulation

The parameters for the genetic algorithm were set as follows: a population of 100 and 300 iterations. The crossover probability ranged from 0.4 to 0.2, and the mutation probability ranged from 0.2 to 0.05, respectively. The size of LOS was (160 m, 72 m, 10 m), with a unit distance of LOV set at 0.4 m, as depicted in Figure 11. In the experiment involving two and three forty-channel lidars at Intersections #1 and #2, the placement of the first (reference) lidar was at (−7.9, −9.1, 2.9, −0.4, 0.5).

The first experiment (S1~S3) was conducted at Intersection #1. Table 3 presents the LO for two forty-channel lidars, three forty-channel lidars, and one one-hundred-and-twenty-eight-channel lidar. Figure 12 displays the placement optimization results for each configuration. In the case of two forty-channel lidars (S1), the placement of the second lidar was determined using the proposed method. Additionally, for three forty-channel lidars (S2), the location of the third lidar was found after the optimal placement in S1 was determined. Figure 12b,c validate the enhancement in measurements achieved by the proposed method. Table 3 shows that the proposed method exhibits performance comparable to that of a 128-channel lidar sensor. The mean and maximum values of LO demonstrate improvements with the proposed method.

The second experiment (S4~S6) was conducted at Intersection #2. Table 4 shows the LO at Intersection #2, and Figure 13 shows the results of placement optimization. As evident from Figure 13b,c, the proposed method improves measurements. Furthermore, it is evident that the proposed method, involving 40-channel multi-lidar placement, demonstrates performance similar to that of the 128-channel lidar, as shown in Figure 13b,c. The average and maximum LO values are either similar to or improved with the proposed method.

#### 5.2.2. Placement and Evaluation in Real Environment

Finally, we placed lidars in the real environment and acquired point cloud data. Among the simulation results for Intersection #1, the optimal placements for two forty-channel lidars (S2) and one one-hundred-and-twenty-eight-channel lidar (S1) were recreated in the real environment. Figure 14 shows the placement of lidars in the actual environment. Three lidars (two forty-channel lidars and one one-hundred-and-twenty-eight-channel lidar) were simultaneously placed, and point clouds were obtained while vehicles and pedestrians were in motion. The point clouds acquired from the two forty-channel lidars and the 128-channel lidar were input into a deep learning-based detection algorithm [28] to compare the positions of detected vehicles and pedestrians. The detection algorithm receives point cloud input from lidar and detects objects such as vehicles and pedestrians based on artificial intelligence, and outputs the type, location, and size of the object in 3D space. Figure 15 shows a detailed scenario, which is as follows.
R1, R2: Point clouds were acquired while vehicles were driven on the road near an intersection. Using a deep learning algorithm, vehicles were detected, and their detection ranges were compared (Figure 15a).R3, R4: Point clouds were acquired as people walked on the sidewalk near the intersection. Pedestrians were detected based on a deep learning algorithm, and the recognized ranges were compared (Figure 15b).R5, R6: Point clouds were acquired as two vehicles were driven on the road near the intersection. While driving, one vehicle was positioned to obscure an area by other vehicles. Using a deep learning algorithm, two vehicles were detected, and the recognition of obscured areas was compared (Figure 15c).

Figure 16 shows the vehicle detection results from the acquired point clouds. Figure 16a shows the results of vehicle detection from the point cloud acquired using the proposed method with two forty-channel lidars. Figure 16b shows the results of vehicle detection from the point cloud acquired with a single 128-channel lidar. Figure 16c,d provide a comparison of the trajectories where vehicles were detected. Table 5 shows the maximum detection position (x, y) and distance. The experiment revealed that the maximum detection position and distance exhibited similar results. However, in areas with sensing blind spots due to buildings, the utilization of low-channel multiple lidars through placement optimization yielded superior detection results. The results of pedestrian detection from the acquired data are shown in Figure 17. Figure 17a shows the results of pedestrian detection from the point cloud acquired using the proposed method with two forty-channel lidars. Figure 17b shows the results of pedestrian detection from the point cloud acquired with a single 128-channel lidar. Figure 17c,d show a comparison of the trajectories where pedestrians were detected. Table 6 presents the maximum detection position (x, y) and distance. The experiment demonstrated that the maximum detection position and distance yielded similar results. Finally, Figure 18 shows the results of detecting an obscured driving vehicle. Figure 18a shows the results of occluded driving vehicle detection from the point cloud acquired using the proposed method with two forty-channel lidars. Figure 18b shows the results of occluded driving vehicle detection from the point cloud acquired with a single 128-channel lidar. Figure 18c,d show a comparison of the trajectories where occluded driving vehicles were detected. Table 7 shows the maximum detection position and distance. Vehicle #1 changes lanes in front of vehicle #2, resulting in temporary obscuration of vehicle #2 when changing lanes. This observation highlights that only occluded vehicles can be detected using the proposed method. It was confirmed that the multi-sensor placement method using the proposed approach is effective in handling occlusions between objects.

The experimental results in the real environment were similar to the simulation results of the proposed method, demonstrating the feasibility of the proposed method. Lidar occupancy was compared by implementing the proposed method in the simulation. Multiple lidars were placed to reduce blind spots at urban intersections, resulting in a higher occupancy rate than high-cost lidar. Multiple lidar placements were replicated in a real environment, and vehicles and pedestrians were detected using point clouds acquired from the lidars. The reproducibility and effectiveness of the proposed method were validated by comparing the detection ranges of the acquired data. In a real environment, multiple lidar placements exhibited a detection range similar to that achieved with a 128-channel lidar. It was evident that placing multiple lidars using the proposed method enhanced measurements, as demonstrated through comparative experiments in a real environment using lidar. For the placement of lidars in an actual urban environment, multiple lidars can be efficiently placed by generating a point cloud map of the environment and applying the proposed method, as demonstrated by experimental results. Our method has showcased that environmental information can be acquired from infrastructures. The acquired environmental information can then be processed to extend the vehicle’s detection range and transmit it to connected vehicles using V2X (Vehicle-to-Infrastructure) communication. This approach not only helps in reducing sensor blind spots in autonomous vehicles, enhancing the safety of other vehicles and pedestrians, but it also holds promise for further research in this domain.

## 6. Conclusions

In this paper, the lidar placement problem was defined as the problem of determining the lidar placement in an urban environment to minimize blind spots and optimize the number of beams reaching the point cloud. To mathematically formalize this problem, a point cloud map was processed and defined as the lidar occupancy space (LOS). Experimental results demonstrated that performance can be enhanced through our lidar placement method. Future work will involve expanding the proposed method to systems utilizing multiple lidar, radar, and cameras, and integrating it with edge–cloud infrastructure and V2X communication technology.

## Figures and Tables

**Figure 1 sensors-23-08808-f001:**
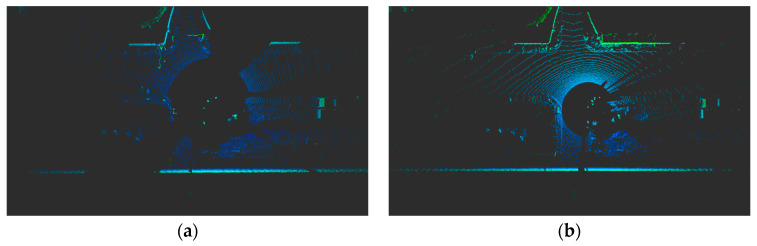
Differences in quality of raw data depending on lidar placement. (**a**) Placement #1. (**b**) Placement #2.

**Figure 2 sensors-23-08808-f002:**
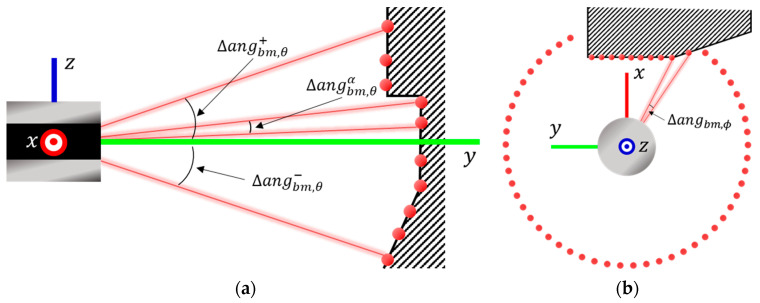
(**a**) Vertical angular resolution of lidar. (**b**) Horizontal angular resolution of lidar.

**Figure 3 sensors-23-08808-f003:**
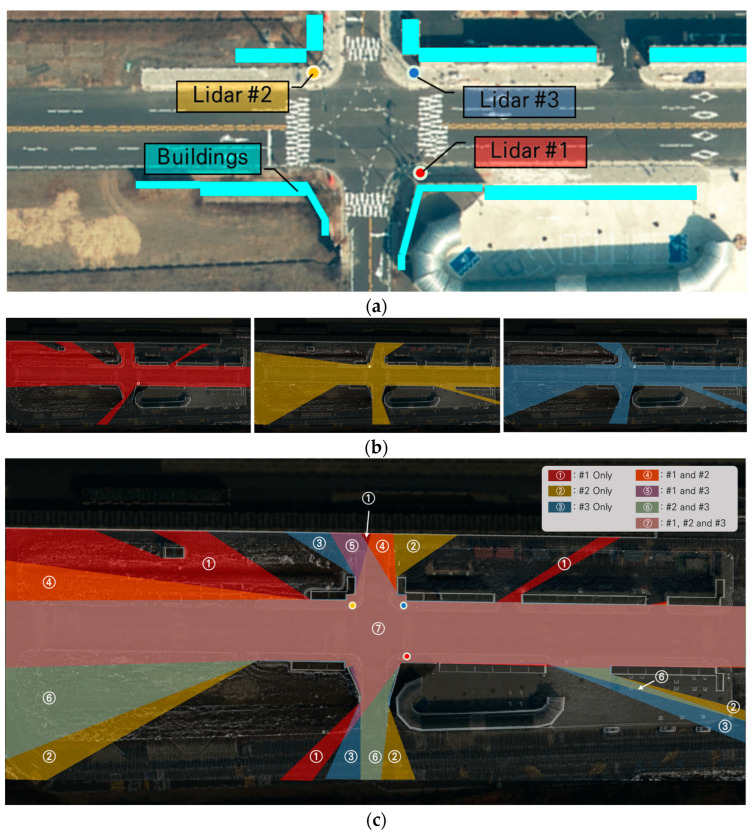
Multiple lidar system in an urban environment. (**a**) Placement of three lidars. (**b**) Coverage of each lidar. (**c**) Total coverage.

**Figure 4 sensors-23-08808-f004:**
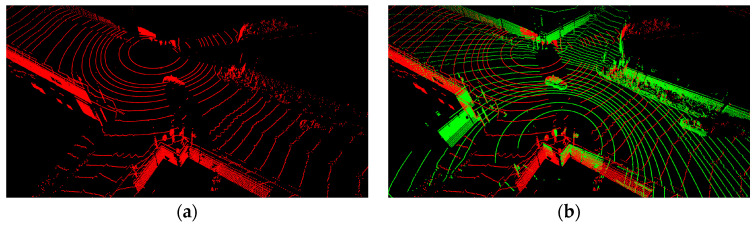
(**a**) Placement of single lidar. (**b**) Placement of dual lidars. (red: point cloud of lidar #1, green: point cloud of lidar #2).

**Figure 5 sensors-23-08808-f005:**
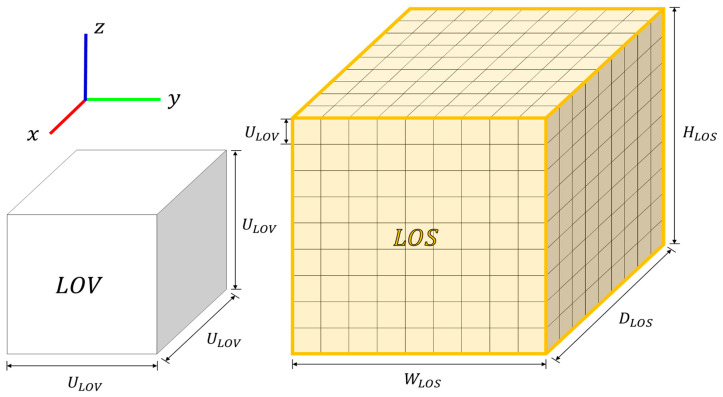
Lidar Occupancy Space (LOS) and Lidar Occupancy Voxel grid (LOV).

**Figure 6 sensors-23-08808-f006:**
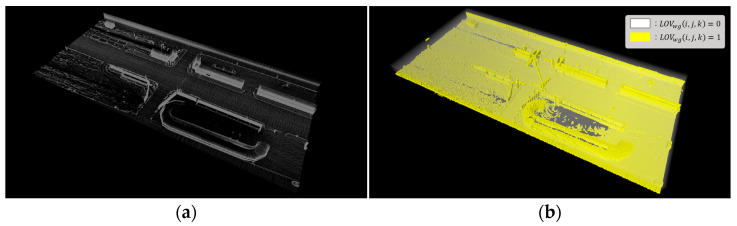
(**a**) Point cloud map. (**b**) Weight for lidar occupancy space.

**Figure 7 sensors-23-08808-f007:**
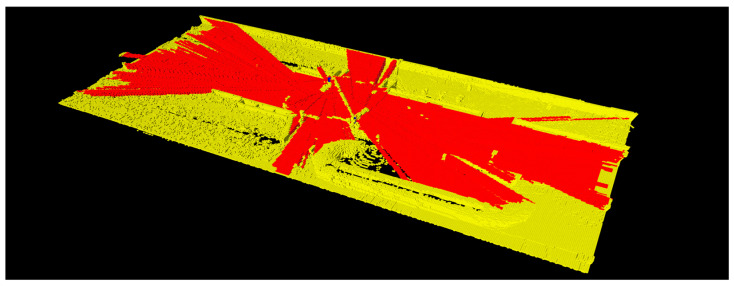
Total lidar occupancy (red voxels).

**Figure 8 sensors-23-08808-f008:**
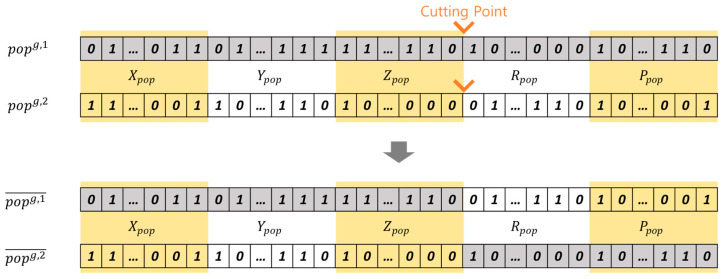
Generation of new chromosomes via the crossover operation.

**Figure 9 sensors-23-08808-f009:**
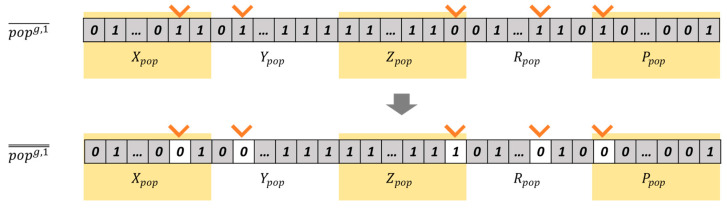
Generation of new chromosomes via the mutation operation.

**Figure 10 sensors-23-08808-f010:**
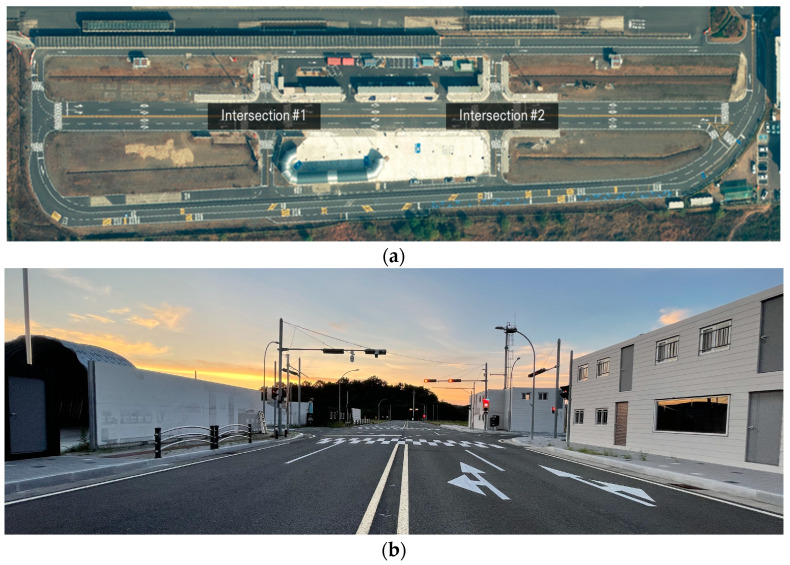
Experimental environment. (**a**) Satellite map. (**b**) 4-way intersection (#1).

**Figure 11 sensors-23-08808-f011:**
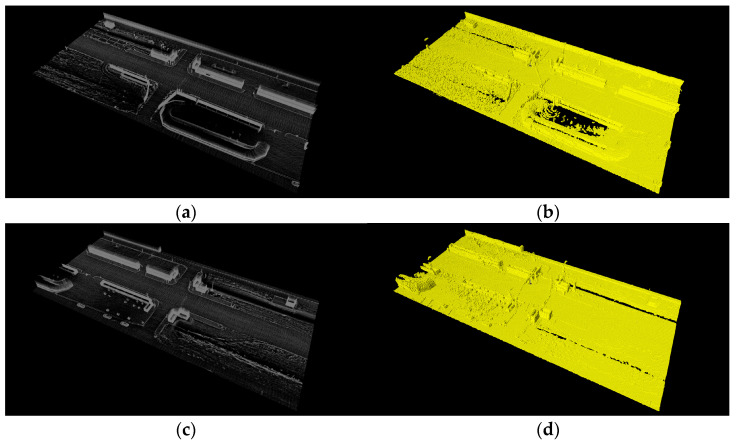
Point cloud map and lidar occupancy space. (**a**,**b**) Intersection #1. (**c**,**d**) Intersection #2.

**Figure 12 sensors-23-08808-f012:**
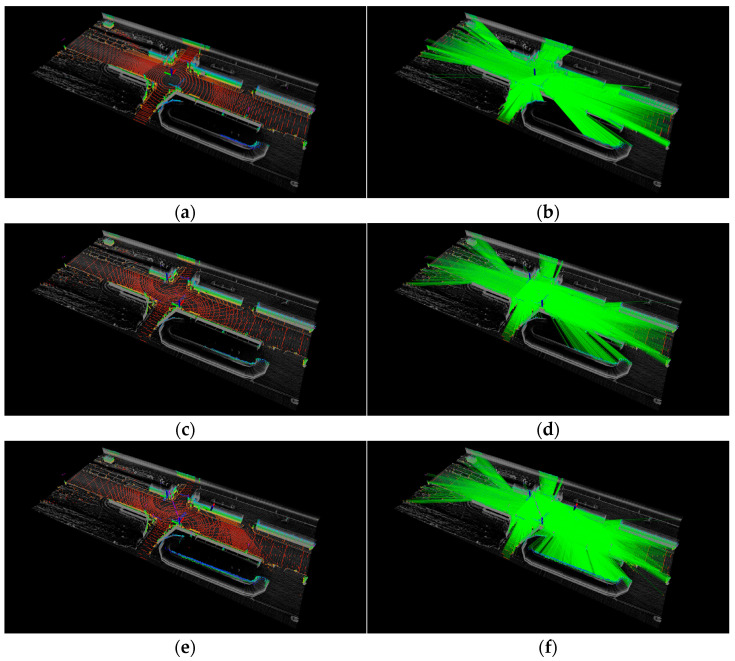
Lidar occupancy space and lidar beams at Intersection #1. (**a**,**b**) One one-hundred-and-twenty-eight-channel lidar (S3). (**c**,**d**) Two forty-channel lidars (S1). (**e**,**f**) Three forty-channel lidars (S2).

**Figure 13 sensors-23-08808-f013:**
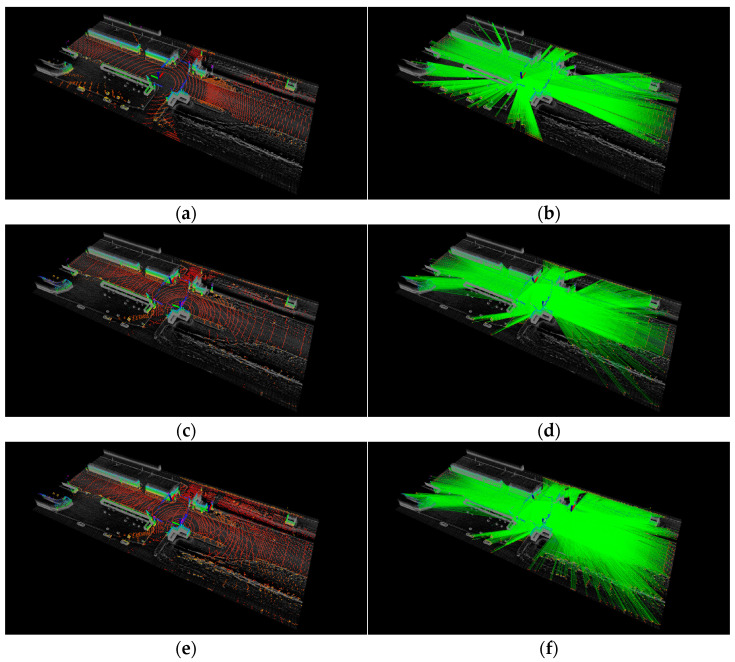
Lidar occupancy space and lidar beams at Intersection #2. (**a**,**b**) One one-hundred-and-twenty-eight-channel lidar (S6). (**c**,**d**) Two forty-channel lidars (S4). (**e**,**f**) Three forty-channel lidars (S5).

**Figure 14 sensors-23-08808-f014:**
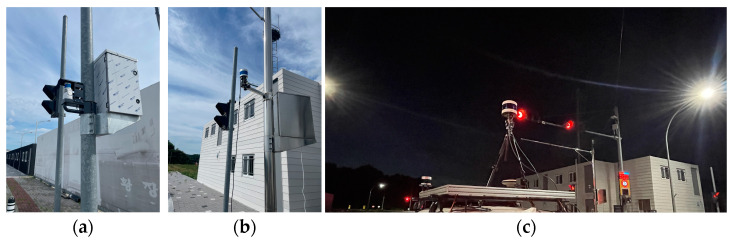
Lidars placed on real environment. (**a**) First 40-channel lidar. (**b**) Second 40-channel lidar. (**c**) The 128-channel lidar.

**Figure 15 sensors-23-08808-f015:**
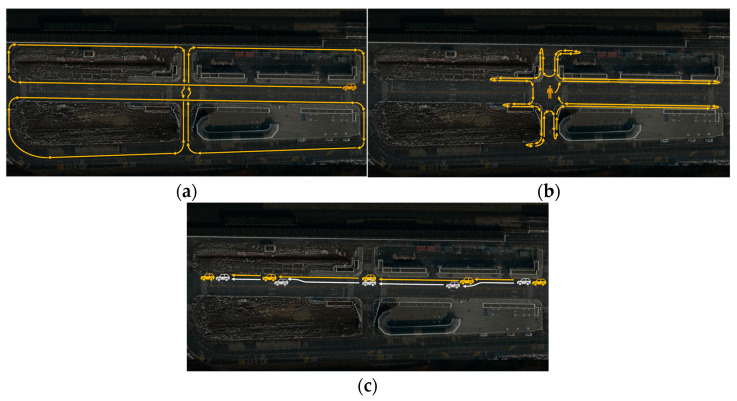
The detailed scenarios. (**a**) Comparison of driving vehicle detection range. (**b**) Comparison of walking pedestrian detection range. (**c**) Comparison of detection range for blocked vehicles.

**Figure 16 sensors-23-08808-f016:**
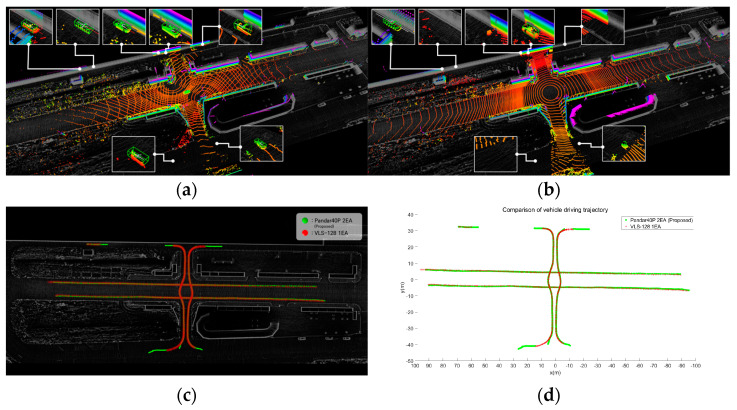
Comparison of vehicle detection range. (**a**) Two forty-channel lidars. (**b**) One one-hundred-and-twenty-eight-channel lidar. (**c**) Visualization of vehicle trajectory. (**d**) Graph of vehicle trajectory.

**Figure 17 sensors-23-08808-f017:**
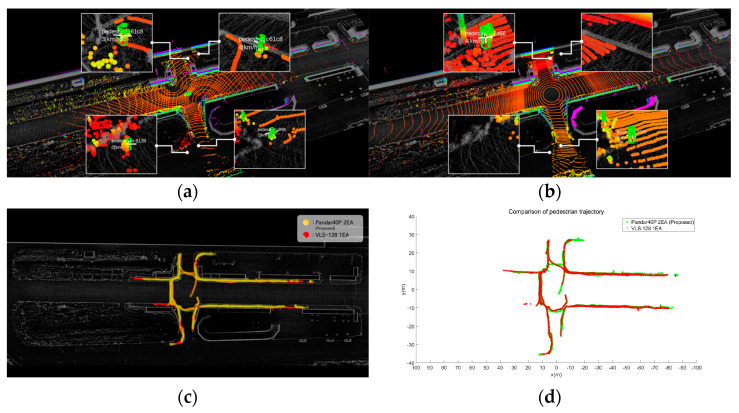
Comparison of pedestrian detection range. (**a**) Two forty-channel lidars. (**b**) One one-hundred-and-twenty-eight-channel lidar. (**c**) Visualization of pedestrian trajectory. (**d**) Graph of pedestrian trajectory.

**Figure 18 sensors-23-08808-f018:**
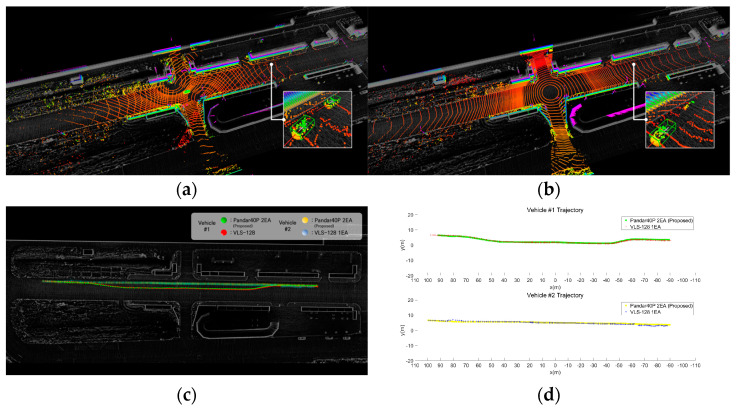
Comparison of occluded vehicle detection range. (**a**) Two forty-channel lidars. (**b**) One one-hundred-and-twenty-eight-channel lidar. (**c**) Visualization of vehicle trajectory. (**d**) Graph of vehicle trajectory.

**Table 1 sensors-23-08808-t001:** Lidar specifications [26,27].

Item	Unit	Specifications	
		Pandar40P	VLS-128
Scan planes	Channel	40	128
Range	m	Up to 200	Up to 245
Range accuracy	cm	±2	±3
FOV (vertical)	Degree	40 (−25 to +15)	40 (−25 to +15)
Resolution (vertical)	Degree	0.33 (non-linear)	0.11 (non-linear)
FOV (horizontal)	Degree	360	360
Resolution (horizontal)	Degree	0.2 (10 Hz), 0.4 (20 Hz)	0.2 (10 Hz), 0.4 (20 Hz)
Frame rate	Hz	10, 20	5 to 20

**Table 2 sensors-23-08808-t002:** Experimental setup.

Config	Environment	Intersection	Sensor		Description
			Model	Number	
S1	Simulator	#1	Pandar40P	2	Lidar placement optimization simulation at Intersection #1
S2				3	
S3			VLS128	1	
S4		#2	Pandar40P	2	Lidar placement optimization simulation at Intersection #2
S5				3	
S6			VLS128	1	
R1	Real	#1	Pandar40P	2	Reproducing optimal placement and comparing vehicle detection ranges
R2			VLS128	1	
R3			Pandar40P	2	Reproducing optimal placement and comparing pedestrian detection ranges
R4			VLS128	1	
R5			Pandar40P	2	Reproducing optimal placement and comparing detection ranges for occluded vehicles
R6			VLS128	1	

**Table 3 sensors-23-08808-t003:** Lidar occupancy according to each parameter (in Intersection #1).

Probability		One One-Hundred-and-Twenty-Eight-Channel Lidar (S3)		Two Forty-Channel Lidars (S1)		Three Forty-Channel Lidars (S2)	
Crossover	Mutation	LO	Best Placement	LO	Best Placement (Second Lidar)	LO	Best Placement (Third Lidar)
0.4	0.2	154,522	$p_{ld}^{1,\;*}$ = (−1.5, 0.3, 2.8, −1.6, −1.0)	138,903	$p_{ld}^{2,\;*}$ = (6.8, 6.0, 4.9, −0.9, 6.0)	174,774	$p_{ld}^{3,\;*}$ = (6.5, −21.1, 3.8, 12.4, −12.4)
	0.1	146,039		157,905		166,702	
	0.05	151,800		147,529		169,019	
	0.02	148,972		152,257		168,000	
0.3	0.2	146,201		137,886		192,507	
	0.1	157,397		141,684		175,029	
	0.05	151,451		140,585		165,479	
	0.02	150,172		141,407		169,823	
0.2	0.2	146,840		145,036		172,465	
	0.1	155,187		147,487		168,328	
	0.05	144,968		148,546		166,546	
	0.02	150,810		152,469		180,816	
0.1	0.2	150,838		143,189		170,464	
	0.1	157,053		**158,553**		169,239	
	0.05	140,229		152,964		172,629	
	0.02	**162,423**		141,022		169,309	
Mean LO		150,931		146,714		171,946	
Max LO		162,423		158,553		192,507	

**Table 4 sensors-23-08808-t004:** Lidar occupancy according to each parameter (in intersection #2).

Probability		One One-Hundred-and-Twenty-Eight-Channel Lidar (S3)		Two Forty-Channel Lidars (S1)		Three Forty-Channel Lidars (S2)	
Crossover	Mutation	LO	Best Placement	LO	Best Placement (Second Lidar)	LO	Best Placement (Third Lidar)
0.4	0.2	144,519	$p_{ld}^{1,\;*}$ = (−6.9, −5.5, 4.7, 1.7, −0.9)	144,376	$p_{ld}^{2,\;*}$ = (6.4, 4.7, 4.1, −2.2, −1.7)	197,441	$p_{ld}^{3,\;*}$ = (7.7, −19.6, 4.9, 3.5, −6.7)
	0.1	143,080		140,789		194,700	
	0.05	140,606		149,373		189,926	
	0.02	146,431		145,034		194,462	
0.3	0.2	146,760		149,409		192,075	
	0.1	143,894		152,930		197,398	
	0.05	151,688		152,350		**203,682**	
	0.02	145,399		**166,184**		191,239	
0.2	0.2	142,365		149,438		191,872	
	0.1	148,304		147,046		192,287	
	0.05	139,525		148,515		194,342	
	0.02	141,120		164,734		187,869	
0.1	0.2	143,224		146,827		187,606	
	0.1	142,970		151,856		174,500	
	0.05	153,015		155,803		174,708	
	0.02	**155,200**		159,735		169,895	
Mean LO		145,506		151,525		189,625	
Max LO		155,200		166,184		203,682	

**Table 5 sensors-23-08808-t005:** Comparison of maximum detection position and distance.

	One One-Hundred-and-Twenty-Eight-Channel Lidar (R2)			Two Forty-Channel Lidars (R1)		
	X	Y	Maximum Distance	X	Y	Maximum Distance
Easting	−95.3	−42.7	95.5	−94.8	−41.0	95.9
Northing	92.2	32.4		95.7	32.4	
Sum	187.5	75.1		190.5	73.4	

**Table 6 sensors-23-08808-t006:** Comparison of maximum detection position and distance.

	One One-Hundred-and-Twenty-Eight-Channel Lidar (R4)			Two Forty-Channel Lidars (R3)		
	X	Y	Maximum Distance	X	Y	Maximum Distance
Easting	−86.4	−35.8	86.7	−85.1	−35.4	85.5
Northing	31.6	28.0		37.7	27.1	
Sum	118.0	63.8		122.8	62.5	

**Table 7 sensors-23-08808-t007:** Comparison of maximum detection position and distance.

		One One-Hundred-and-Twenty-Eight-Channel Lidar (R6)			Two Forty-Channel Lidars (R5)		
		X	Y	Maximum Distance	X	Y	Maximum Distance
Vehicle #1	Easting	−89.2	1.1	91.8	−90.3	0.9	97.8
	Northing	91.6	6.4		97.6	6.5	
	Sum	180.8	5.3		187.9	5.6	
Vehicle #2	Easting	−89.9	3.6	99.7	−87.8	2.7	99.2
	Northing	99.5	6.6		99.1	6.8	
	Sum	189.4	3.0		186.9	4.1	

## Data Availability

Data will be made available on request.

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
