# Peer review of "Placement Method of Multiple Lidars for Roadside Infrastructure in Urban Environments"

_sensors, 2023, doi:10.3390/s23218808_

Round 1

Reviewer 1 Report

Comments and Suggestions for Authors

After reading this paper, I think this paper is complete and contains all the necessary components. The structure is reasonable and logical. This manuscript introduces voxel grids and then applies genetic algorithms to solve the optimization problem of sensor position placement. This submission seems interesting. However, even though the methodological approach is explained clearly, there are still several issues that need addressing, as my comments below for details:

1. The authors should add discussion about the contribution in the paper, because some similar works have been published in the recent years. I could not find new contribution clearly in the manuscript.

2. Although the authors labeled what the images represent, the specific information of each sub-figure lacks the necessary textual explanation that is essential for others to understand this experiment.

3. It is noted that your manuscript needs careful editing by someone with expertise in technical English editing paying particular attention to English grammar, spelling, and sentence structure so that the goals and results of the study are clear to the reader.

4. In general, there is a lack of explanation of methods used in the study. Furthermore, an explanation of why the authors did these various experiments should be provided.

5. The literature review should be improved by citing more relevant studies. Just list several as follows.

TrajMatch: Toward Automatic Spatio-Temporal Calibration for Roadside LiDARs Through Trajectory Matching

Traffic Sign Based Point Cloud Data Registration with Roadside LiDARs in Complex Traffic Environments

Research on carbon emissions of public bikes based on the life cycle theory

6. Some minor modification suggestions:

(1) The formatting of references needs to be kept consistent and in line with the requirements of academic journals, with some references missing information such as page numbers;

(2) Punctuation is missing from the titles of some figures, such as Figure 1.

(3) The title of Figure 9 appears at the top of the figure, which is inconsistent with the placement of the titles of the other figures. It is hoped that the authors will double-check the relevant formatting.

Comments on the Quality of English Language

need to be improved

Reviewer 2 Report

Comments and Suggestions for Authors

The paper corresponds to new technology dedecated to monitoring the enviroment of road infrastrucure using Lidars are multiply lidars. The problems related to positioning the lidars in the presented project are defined with respect to mathematical criteria. (especially optimisation).

All formulas  are clear and correct from mathematical point of viev. The charts, figures and tables are correct and readible.

The problem is new (lida rapplications in road enviroment), now the authors described the experiments, but the modelling (with simulation and optimisation results) give the positive aspects for future applications.

Reviewer 3 Report

Comments and Suggestions for Authors

Very interesting and important study which considers a very practical problem faced when using lidars or similar technology. The study is also well structured and shows the math and techniques used to validate the theory and find the ideal position for the lidar. What I think is missing is a more extensive background and comparisons showing how not having the ideal location can lead to issues in real-world applications. Additionally, the discussions following the results are very short and could be expanded on to better explain the implications of the results and a way forward for practical development and further study. Overall, it is a good piece of work. 

Comments on the Quality of English Language

English needs moderate corrections for grammar.

Round 2

Reviewer 1 Report

Comments and Suggestions for Authors

The authors have dealt with most of my comments. However, I suggest that the authors further improve the paper, especially for the third and fifth comments.

Comments on the Quality of English Language

Good
